# Mechanotransduction and Skeletal Muscle Atrophy: The Interplay Between Focal Adhesions and Oxidative Stress

**DOI:** 10.3390/ijms26062802

**Published:** 2025-03-20

**Authors:** Khaled Y. Kamal, Marina Trombetta-Lima

**Affiliations:** 1Department of Kinesiology, Iowa State University, Ames, IA 50011, USA; 2Department of Pharmaceutical Technology and Biopharmacy, Groningen Research Institute of Pharmacy, University of Groningen, 9700 Groningen, The Netherlands; m.trombetta.lima@rug.nl

**Keywords:** skeletal muscles, focal adhesions, oxidative stress, redox biology, mechanotransduction, NADPH oxidases, muscle atrophy, therapeutic interventions

## Abstract

Mechanical unloading leads to profound musculoskeletal degeneration, muscle wasting, and weakness. Understanding the specific signaling pathways involved is essential for uncovering effective interventions. This review provides new perspectives on mechanotransduction pathways, focusing on the critical roles of focal adhesions (FAs) and oxidative stress in skeletal muscle atrophy under mechanical unloading. As pivotal mechanosensors, FAs integrate mechanical and biochemical signals to sustain muscle structural integrity. When disrupted, these complexes impair force transmission, activating proteolytic pathways (e.g., ubiquitin–proteasome system) that accelerate atrophy. Oxidative stress, driven by mitochondrial dysfunction and NADPH oxidase-2 (NOX2) hyperactivation, exacerbates muscle degeneration through excessive reactive oxygen species (ROS) production, impaired repair mechanisms, and dysregulated redox signaling. The interplay between FA dysfunction and oxidative stress underscores the complexity of muscle atrophy pathogenesis: FA destabilization heightens oxidative damage, while ROS overproduction further disrupts FA integrity, creating a self-amplifying vicious cycle. Therapeutic strategies, such as NOX2 inhibitors, mitochondrial-targeted antioxidants, and FAK-activating compounds, promise to mitigate muscle atrophy by preserving mechanotransduction signaling and restoring redox balance. By elucidating these pathways, this review advances the understanding of muscle degeneration during unloading and identifies promising synergistic therapeutic targets, emphasizing the need for combinatorial approaches to disrupt the FA-ROS feedback loop.

## 1. Introduction

Significant efforts have been made to understand the molecular and cellular mechanisms underlying disuse-induced skeletal muscle atrophy, and several signaling pathways have been studied. However, key gaps remain in understanding the regulatory mechanisms driving this process and their functional significance. Skeletal muscles are dynamic, multifaceted tissue responsible for producing body movement, contributing to joint and bone protection, acting as a glucose sink, and regulating systemic energy metabolism. Skeletal muscles adapt and remodel in response to changes in mechanical loading, nutrient availability, and energy demands. Mechanical overloading induces hypertrophy, enhancing muscle mass and function, while mechanical unloading leads to atrophy, characterized by increased protein degradation and reduced protein synthesis. A possible teleological explanation for atrophy is its role as a protective mechanism in extreme disuse due to injury or disease. It would require an increased demand for amino acids and proteins [1,2]. This response helps conserve resources for repairing damaged or infected tissues and maintaining homeostasis [2]. Despite extensive studies on disuse-induced atrophy [3], critical questions remain regarding the functional roles of specific signaling pathways and their therapeutic potential. Addressing these questions is vital for designing effective interventions to mitigate muscle atrophy in clinical and physiological settings [4,5,6,7].

The complexity of skeletal muscles extends beyond movement and metabolism. These tissues play pivotal roles in maintaining systemic health, helping to mitigate metabolic diseases, and maintaining energy balance [8,9]. The prevalence of conditions leading to muscle atrophy, including aging, spaceflight, and prolonged immobilization, underscores the urgency of advancing our understanding of these processes. Recent research has revealed critical insights into how mechanical unloading alters cellular signaling pathways, offering potential therapeutic targets. These highlight the potential for targeted therapeutic strategies that could mitigate the effects of mechanical unloading on muscle tissue, thus benefiting a wide range of patients, from astronauts to the elderly confined to prolonged bed rest [10,11]. Skeletal muscles respond and remodel to mechanical, nutrient, and energy-sensing and stress changes. This adaptation can manifest as hypertrophy in response to chronic overloading or atrophy under conditions of mechanical unloading, such as hindlimb unloading, spaceflight, or bedrest.

This review focuses on the pivotal roles of focal adhesions and oxidative stress within the broader mechanotransduction pathways. Discussing these mechanisms, we aim to provide new perspectives on their critical impacts on muscle health and highlight promising therapeutic strategies. Focal adhesions serve as specialized sites for cytoskeletal connections to the extracellular matrix (ECM), functioning as structural links and signaling hubs. Their dysregulation has been implicated in muscular dystrophy, muscle atrophy, and impaired force transmission. Oxidative stress, characterized by an imbalance between ROS production and antioxidant defense, exacerbates muscle dysfunction, providing another layer of complexity [12,13].

By addressing these pathways, this review aims to advance the understanding of mechanotransduction and its role in muscle atrophy during mechanical unloading, identifying novel therapeutic targets and highlighting the pressing need for continued research.

## 2. Mechanotransduction in Skeletal Muscle

Mechanotransduction is the process by which cells convert mechanical signals into biochemical signals. This process is fundamental for tissue function, adaptation, and homeostasis in skeletal muscle, especially under varying mechanical loads [14]. Mechanical signals arise from muscle contractions, external forces, and changes in tissue architecture (Table 1), which summarizes key mechanosensory proteins and their roles in regulating these responses. These signals are detected and processed by specialized “mechanosensors”, which include stretch-activated ion channels at the cell membrane [15], transmembrane adhesion receptors that link the cytoskeleton to the ECM, and dystrophin–glycoprotein complexes [16,17,18]. Additionally, sarcomeric proteins like actin and myosin [19] and specific cell-surface receptors [20] contribute to this complex signaling network.

Recent research highlights distinct mechanotransduction mechanisms across skeletal muscle cell types. Satellite cells, the quiescent muscle stem cells, rely on integrin-mediated adhesion to the ECM for activation and proliferation, with focal adhesion kinase (FAK) signaling critical for their mechanoresponsiveness [21]. Upon differentiation into myoblasts, cells exhibit heightened sensitivity to substrate stiffness via stretch-activated ion channels (e.g., Piezo1) and cytoskeletal remodeling, which guide migration and alignment [20,22]. In contrast, mature myotubes depend on costamere-linked mechanosensors, such as the dystrophin–glycoprotein complex (DGC) and YAP/TAZ pathways, to transduce contractile forces into anabolic signals, maintaining structural integrity [23,24]. Disruptions in these pathways—such as FAK inactivation in satellite cells or DGC dysfunction in myotubes—impair regeneration and exacerbate atrophy [6,25]. These stage-specific mechanotransduction differences underscore the need for targeted therapies to address cellular heterogeneity during muscle adaptation.

The interplay of mechanosensors facilitates the integration of mechanical stimuli into coordinated cellular response, enabling the skeletal muscle to maintain its structural and functional integrity during mechanical loading and unloading. Mechanotransduction pathways overlap significantly with those involved in nutrient sensing [26,27], energy homeostasis [28,29], and oxidative-stress regulation [30,31]. These include key pathways regulated by insulin and insulin-like growth factor I (IGF-I) [32], adenosine monophosphate-activated protein kinase (AMPK) [33], ROS [34], and prostaglandins (PGs) [35]. This overlap underscored the interconnectedness of metabolic and mechanical signaling in muscle tissue.

Emerging evidence highlights that disruption in mechanotransduction pathways can contribute to muscle dysfunction, including atrophy and impaired regeneration. Stretch-activated ion channels, for example, serve as early responders to mechanical perturbations, triggering downstream signaling cascades that regulate gene expression, protein synthesis, and cellular adaptation [20]. Similarly, integrins, transmembrane receptors linking the ECM to the actin cytoskeleton, play dual roles in mechanosensing and signal transduction. They mediate bidirectional communication between the cell and its external environment through ‘inside-out’ and ‘outside-in’ signaling mechanisms [36,37].

Mechanical signals also regulate energy metabolism and intracellular signaling through mechanotransduction pathways. These processes are closely tied to the skeletal muscle’s ability to adapt to varying demands, such as hypertrophy during overloading and atrophy during unloading. Integrating mechanical, nutrient, and energy sensing ensures muscle tissue responds appropriately to external stimuli, maintaining muscle mass and function even under physiological stress. Recent advances in mechanotransduction research have unveiled novel therapeutic targets for mitigating skeletal muscle atrophy. For instance, modu-lating integrin-associated focal adhesion complexes and tagging key signing molecules, such as focal adhesion kinase (FAK), could offer new strategies to preserve muscle in-tegrity under mechanical unloading. By elucidating the molecular basis of mecha-notransduction, these findings provide a foundation for developing interventions to address muscle dysfunction in clinical and extreme environments.

**Table 1 ijms-26-02802-t001:** Key mechanotransduction proteins in skeletal muscles and their functions.

Mechanosensory Protein	Function	Mechanisms	References
Neuronal Nitric Oxide Synthase (nNOS)	Regulates blood flow, insulin-induced glucose uptake, satellite-cell activation, and muscle regeneration.	Localizes to the sarcolemma, interacts with the dystrophin–glycoprotein complex, and produces nitric oxide (NO), modulating vasodilation and glucose uptake. Disruption leads to impaired repair and increased inflammation.	[38,39,40,41,42,43]
Perlecan	Acts as a mechanosensory protein regulating metabolism, muscle growth, and repair.	Binds ECM components, interacts with growth factors, and modulates hypertrophy/atrophy pathways. Promotes nNOS delocalization during atrophy, enhancing protein degradation.	[44,45,46,47]
Focal-Adhesion Complex	Links actin cytoskeleton to the ECM, transmitting mechanical signals for gene expression and cellular adaptation.	Composed of integrins, talin, vinculin, and FAK, mediating biochemical responses to mechanical stimuli. Dysfunction leads to disrupted force transmission and impaired regeneration.	[48,49,50]
Dystrophin–Glycoprotein Complex (DGC)	Maintains sarcolemma integrity and transmits contractile forces.	Anchors actin cytoskeleton to the ECM, interacting with dystroglycans and sarcoglycans. Loss of function leads to sarcolemma instability.	[51,52,53]
Dysferlin	Facilitates membrane repair, which is critical for muscle recovery post injury.	Involved in vesicle fusion and resealing, interacting with caveolin-3 and annexins. Deficiency results in impaired repair and chronic inflammation.	[54,55,56,57]
Integrins	Mediates cell–ECM adhesion and transmits mechanical signals.	Binds ECM proteins like fibronectin/collagen and activates intracellular signaling via FAK. Contributes to stress fiber formation and mechanosensing.	[58,59,60,61,62]
Talin	Links integrins to the actin cytoskeleton and supports focal adhesion signaling.	Binds integrin cytoplasmic tails, recruiting vinculin and other focal adhesion proteins. Impairment weakens focal-adhesion strength, leading to atrophy.	[36,63,64,65]
Vinculin	Reinforces cell–ECM adhesion and regulates cell motility.	Interacts with talin and actin, stabilizing focal adhesions and modulating actin dynamics. Essential for maintaining structural integrity during stress.	[66,67,68,69,70]
Yes-associated protein (YAP) and transcriptional coactivator with PDZ-binding motif (TAZ) (YAP/TAZ)	Transcriptional co-activators modulate gene expression via mechanical signals.	Translocate to the nucleus under mechanical stimuli, interacting with transcription factors. Critical for stress adaptation, dysregulation promotes atrophy.	[24,71,72,73,74]
Piezo 1/2	Mechanosensitive ion channels regulate the cellular response to mechanical stimuli.	Open under force to mediate ion influx, triggering downstream signaling pathways. Hyperactivation disrupts homeostasis and promotes oxidative stress.	[22,75,76,77]

## 3. Focal Adhesions as Mechanosensors

Focal adhesions are integral components of cellular mechanosensing machinery acting as critical links between the ECM and the cytoskeleton. These dynamic complexes facilitate the transmission of mechanical forces into biochemical signals, crucial for maintaining cellular integrity and regulating various processes, including muscle contraction, adaptation, and repair. Focal adhesions consist of numerous proteins, such as integrins, focal adhesion kinase (FAK), talin, vinculin, dysferlin, and perlecan, which work synergistically to mediate structural and signal functions (Figure 1) [78,79,80].

### 3.1. Key Molecular Components in Focal Adhesions

#### 3.1.1. Integrins

Integrins play a crucial role as mechanotransducers that transmit force [81]. Integrins bind to ECM proteins through specific domains, such as the RGD (Arginine–Glycine–Aspartate) motif in fibronectin and laminin-specific domains in the muscle basement membrane, thereby facilitating force transmission (particularly integrin α7β1 binding to laminin is crucial in skeletal muscle) [82,83]. The functional integrin unit consists of a paired α and β subunit and acts independently or in conjunction with cell-membrane proteoglycans to modulate cell adhesion synergistically [84,85,86]. Therefore, the orchestrated expression of different integrin subunits and their associated partner proteins is crucial for muscle differentiation and functionality. The dysregulation of integrin expression or activity is linked to many muscular dysfunctions. For instance, integrin α7 dysfunction is related to congenital and late-onset conditions, such as congenital muscular dystrophy, Duchenne muscular dystrophy, and late-onset cardiac dysfunction [87,88,89,90].

While the DGC primarily provides a structural link between the cytoskeleton and the ECM, associating with the intermediate filament system and providing mechanical stability to the cellular membrane, integrins play a bidirectional role in the communication between the cell and its microenvironment [53,62]. On the one hand, integrins are transported to the cellular membrane in an inactive conformation and rely on intracellular signaling—through the interaction of their β subunit intracellular tail with talin and kindlin—to expose its extracellular ECM binding sites, influencing how the cell perceives its external microenvironment (‘inside-out’ signaling) [36,37]. On the other hand, integrin interaction with a diverse range of ECM ligands functions as mechanosensory machinery, regulating cellular processes, such as stress-fiber formation and proliferation, allowing for force transmission across the muscle-fiber membrane and modulating cellular responses to the microenvironment (‘outside-in’ signaling) [91,92].

The interaction between the DGC and integrin-based focal adhesions through cytoskeletal proteins such as talin and vinculin provides a robust mechanical linkage at the costameres. These specialized regions play a pivotal role in the force transmission from sarcomeres to the sarcolemma, essential for muscle contraction and growth [79,93,94,95,96].

#### 3.1.2. Focal Adhesion Kinase (FAK)

FAK is integral to skeletal muscle mechanotransduction, serving as a nexus for integrating mechanical cues into biochemical signals. FAK activation is initiated by integrin clustering at focal adhesions during mechanical stress, triggering autophosphorylation at Tyr397 and recruiting Src family kinases, which phosphorylate downstream targets such as p130Cas and paxillin, thus activating anabolic pathways (e.g., PI3K/Akt/mTOR signaling) and inhibiting catabolic signaling (FoxO-mediated proteolysis) [49,80]. Structurally, FAK’s FERM domain interacts with integrins and growth-factor receptors, while the FAT domain anchors FAK to focal adhesions, enabling force-dependent conformational changes critical for its mechanosensory function [97]. Under unloading or disuse, FAK becomes inactive, disrupting costamere integrity and exacerbating muscle atrophy through the activation of NF-κB and the ubiquitin–proteasome system (UPS) [98]. Additionally, FAK coordinates with nNOS, where NOX2-derived ROS destabilizes FAK during muscle disuse, compromising force transmission [6,98]. The therapeutic activation of FAK is considered promising for preventing disuse-induced muscle wasting and promoting regeneration through YAP/TAZ nuclear signaling [24,98].

#### 3.1.3. Talin and Vinculin

Talin and vinculin collaborate as crucial mechanosensitive adapters linking integrins to the actin cytoskeleton. Talin, a dimeric protein, binds integrin β-subunit cytoplasmic tails via its FERM domain and undergoes force-induced unfolding of rod domains to expose cryptic vinculin-binding sites [36,65]. Vinculin recruitment strengthens the integrin–cytoskeletal linkage, creating a “molecular clutch” that transmits mechanical forces to intracellular structures, influencing gene expression and cytoskeletal remodeling [66,68]. During unloading or muscle inactivity, calpain-mediated degradation of talin and delocalization of vinculin disrupt focal adhesion assembly, reducing force transmission and activating atrophic signaling pathways [99]. Vinculin-null myofibers show reduced stiffness and increased vulnerability to contraction-induced injury, underscoring the importance of talin–vinculin integrity for muscle protection [70]. Pharmacological and exercise-based approaches that restore talin–vinculin interactions offer therapeutic potential by enhancing focal adhesion stability under mechanical stress [98].

#### 3.1.4. Dysferlin

Dysferlin, a Ca^2^⁺-sensitive protein, is critical for skeletal muscle membrane repair and mechanotransduction. It is involved in sarcolemmal repair and indirectly interfaces with focal adhesion signaling by associating with integrins and FAK, thus contributing to the mechanotransduction pathway. Dysferlin’s primary role is mediating vesicle fusion at injury sites via interactions with annexins and MG53, restoring sarcolemmal integrity [54,57]. Beyond repair, dysferlin modulates FA signaling by associating with β1-integrins and FAK, influencing mechanoadaptive responses [100]. Dysferlin deficiency (e.g., in dysferlinopathy) impairs FAK activation, leading to aberrant ROS production via NOX2 and disrupted nNOS localization, which exacerbates atrophy [6,101]. In unloading models, dysferlin-null mice exhibit accelerated muscle loss and fibrosis due to unresolved membrane damage and defective satellite-cell activation [102,103]. Dysferlin also interacts with caveolin-3, linking membrane repair to lipid raft signaling; its loss disrupts IGF-1 receptor trafficking, impairing hypertrophy [88]. In Duchenne muscular dystrophy, secondary dysferlin dysfunction amplifies pathology by hindering compensatory repair mechanisms, leading to chronic inflammation and fibrotic replacement [51]. Although not a classical focal-adhesion protein, dysferlin integrates mechanotransduction signals via its association with integrins and FAK, serving as an auxiliary mechanosensor involved in membrane repair and adaptive signaling. Therapeutic strategies targeting dysferlin function, such as gene therapy and membrane-stabilizing agents (e.g., poloxamers), show promise for improving muscle integrity and mechanotransduction [57].

#### 3.1.5. Perlecan

Perlecan is a heparan sulfate proteoglycan in the extracellular matrix of skeletal muscle tissue. It is a mechanosensitive protein interacting with integrins and other membrane-associated adhesion proteins. It can detect mechanical forces acting on the muscle and transmit this information to the cells [45]. Mechanical forces exerted on the ECM are sensed by perlecan, which then communicates these signals to muscle fibers via interactions with cell-surface integrins and dystroglycan, thereby modulating intracellular signaling pathways [104]. Perlecan’s mechanosensitivity significantly influences muscle-fiber composition, affecting hypertrophy and atrophy responses [104,105]. Notably, perlecan mediates mechanosensitive signaling by regulating nNOS localization; mechanical unloading or denervation leads to perlecan-dependent nNOS displacement from the sarcolemma, subsequently activating FoxO transcription factors and the ubiquitin–proteasome system (UPS), driving muscle atrophy [106,107]. Thus, perlecan contributes to muscle adaptation to mechanical forces and represents an emerging therapeutic target for muscle wasting disorders.

### 3.2. Functional Integration of Focal Adhesion Components

Focal adhesions operate through integrin-based signaling to ensure robust force transmission and cellular adaptation. Central to their function is the anchoring of the actin cytoskeleton to the extracellular matrix (ECM), a process mediated by talin and vinculin, which mechanically couple integrins to intracellular filaments, enabling bidirectional force transmission [71,73]. The interaction between integrin-based focal adhesions and the DGC provides a cohesive mechanical network at costameres, supporting sarcolemma integrity during muscle contraction and growth [94,95]. ECM stiffness critically regulates this interplay, as increased matrix rigidity enhances integrin clustering and FAK activation, promoting myoblast differentiation, while soft substrates favor satellite-cell quiescence [36,108]. Under the conditions of mechanical unloading, including muscular dystrophy, atrophy, and impaired regeneration, focal adhesion dynamics are altered, leading to disrupted signal transmission and increased susceptibility to muscle damage [78,99,109,110,111].

Recent research has highlighted the role of FAK in mediating mechanotransduction pathways. FAK, a central signaling molecule activated by mechanical stress, regulates cell survival, proliferation, and differentiation processes. During muscle cell differentiation, FAK phosphorylates downstream targets like paxillin and ERK, driving myoblast fusion into multinucleated myotubes [100,112]. The dysregulation of FAK signaling has been implicated in muscle atrophy, making it a potential therapeutic target [21,78,97,98]. Similarly, vinculin, an actin-binding protein, reinforces cell–ECM adhesion by strengthening focal adhesions and modulating actin dynamics, further emphasizing the mechanosensitive nature of these complexes [66,68]. Vinculin’s interaction with talin ensures cytoskeletal anchoring under mechanical strain, stabilizing focal adhesions during muscle contraction [65,66]. The adaptability of focal adhesions to mechanical cues is essential for skeletal muscle function. These structures integrate signals from ECM and cytoskeleton, allowing muscle cells to respond dynamically to changes in mechanical load. For example, under mechanical unloading, focal-adhesion composition and function are altered, leading to disrupted force transmission and increased susceptibility to muscle damage [99]. This disruption underscores the importance of focal adhesions in maintaining muscle health and their potential as targets for therapeutic interventions. In addition to their structural role, focal adhesions serve as platforms for intracellular signaling, influencing muscle adaptation and repair pathways.

In addition to their structural role, focal adhesions serve as biochemical hubs for muscle differentiation. YAP/TAZ transcriptional co-activators, regulated by ECM stiffness and cytoskeletal tension, translocate to the nucleus in stiff environments, activating pro-myogenic genes (e.g., MyoD) and suppressing adipogenic pathways, thereby steering myoblasts toward terminal differentiation [24,113]. Emerging technologies, such as super-resolution microscopy and molecular modeling, have provided novel insights into the real-time dynamics of focal adhesions, shedding light on their role in mechanosensing and cellular adaptation [80].

In summary, focal adhesions are indispensable mechanosensors in skeletal muscle, integrating mechanical and biochemical signals to maintain muscle function and integrity. Understanding the molecular mechanisms and interactions within these complexes, including integrins, talin, vinculin, dysferlin, and perlecan, offers valuable insights into therapeutic strategies for mitigating muscle atrophy and dysfunction associated with mechanical stress.

## 4. Oxidative Stress in Muscle Atrophy

Oxidative stress is a key driver of skeletal muscle atrophy, resulting from an imbalance between reactive oxygen species (ROS) production and the antioxidant defense mechanism. ROS are byproducts of aerobic metabolism, primarily generated by the mitochondrial electron transport chain, NADPH oxidases (NOX), and xanthin oxidase, making them essential signaling molecules and contributing to oxidative damage [108,112,114,115]. ROS production increases significantly during prolonged muscle inactivity, aging, or disease, causing oxidative modifications to proteins, lipids, and nucleic acids, ultimately disrupting cellular homeostasis and accelerating muscle atrophy. This imbalance underpins the progression of muscle atrophy through increased protein degradation, impaired synthesis, and reduced regenerative capacity [113,116,117,118,119,120,121]. In addition, excessive ROS disrupts insulin signaling pathways, fostering insulin resistance, a common metabolic dysfunction that further impairs muscle health [12,13].

Muscle damage, oxidative stress, and associated inflammation negatively impact skeletal muscle by primarily reducing protein synthesis, leading to muscle wasting. Mitochondria are a significant source of ROS in inactive skeletal muscle fibers. At the same time, prolonged muscle inactivity also stimulates ROS production through cytosolic mechanisms, such as xanthine oxidase and NADPH oxidase activation [122,123,124,125,126,127,128]. Initially identified in immune cells like neutrophils and macrophages, NADPH oxidases are recognized as key players in skeletal muscle ROS production during muscle contraction and disuse. NOX enzymes are a complex of six subunits: gp91phox (NOX2), p22phox, p47phox, p67phox, p40phox, and Rac. Among them, NOX2 is the primary functional unit across various cell types, contributing to calcium regulation in muscle cells and mechanotransduction signaling [6,129,130].

The mitochondrial production of ROS plays a central role in triggering these cat-abolic processes, exacerbating muscle atrophy during periods of disuse such as hindlimb unloading [127,131,132,133,134,135]. While mitochondrial ROS production is well studied, cytosolic ROS sources such as xanthine oxidase’s specific contribution remain unclear. ROS also negatively impacts skeletal muscle regeneration by impairing the function of satellite cells. This dysfunction leads to labile iron accumulation, increased lipogenesis, and a decrease in crucial antioxidant proteins like glutathione peroxidase 4 (Gpx4) and nuclear factor erythroid 2–related factor 2 (Nrf2), ultimately hindering the muscle’s ability to recover from injury and contributing to the pathogenesis of sarcopenia [136,137]. On the other hand, ROS facilitates the macrophage-mediated clearance of cellular debris, a process essential for effective muscle regeneration, highlighting its dual role in muscle health [138].

### 4.1. NADPH Oxidase Family as Key Mediator

The NADPH oxidase family comprises membrane-bound oxidoreductase complexes, including skeletal muscle Nox2, Nox4, and Duox2-dependent isoforms. NADPH Oxidase-2 (NOX2) plays a dual role in skeletal muscle physiology and pathology. Under normal conditions, it regulates signaling pathways for muscle adaptation, including glucose uptake and contractile activity. However, excessive NOX2 activation leads to pathological outcomes, such as increased ROS production, oxidative damage, and impaired muscle-repair mechanisms. The NADPH oxidase family comprises membrane-bound oxidoreductase complexes, including skeletal muscle NOX2 and NOX4, as the predominant isoforms in skeletal muscle (Figure 2). These enzymes play an essential role in ROS production, acting as signaling mediators under physiological conditions but exacerbating oxidative damage when dysregulated [139,140].

Nox2 becomes hyper-responsive to stretching in dystrophic conditions, such as muscular dystrophies, causing excess oxidative stress and contributing to pathology in mice [141]. For example, Lawler’s laboratory demonstrated that Nox2 is the primary source of ROS during unloading-induced atrophy, where hyperactivation disrupts the localization of neuronal nitric oxide synthase (nNOS) [6,130]. Also, Nox2 plays a role in the response of the heart and skeletal muscle to mechanical stretch. Nox2 was upregulated in skeletal muscle in response to stretch, and this upregulation was necessary for activating signaling pathways that lead to muscle growth [142].

Interestingly, NADPH oxidases positively regulate metabolism and contractile activity at low to moderate levels. For instance, nNOSµ and Nox4 collaborate to produce peroxynitrite (OONO−), facilitating the initial stages of skeletal muscle hypertrophy [143,144]. However, elevated levels of Nox2 lead to nNOSµ translocation into the cytosol, shifting toward atrophic signaling pathways [6]. This mislocalization reduces nitric oxide bioavailability, further impairing muscle homeostasis. Inhibitors of Nox2, such as peptide-based therapies, have shown efficacy in mitigating unloading-induced skeletal muscle atrophy by preserving mechanotransduction signaling and reducing ROS levels [6]. Additionally, Nox2 activation is linked to ligand binding at the angiotensin II receptor type 1 (AT1R), forming a mechanosensitive signaling complex that exacerbates oxidative damage and muscle atrophy [130,145,146].

In conclusion, Nox2 represents a double-edged sword in skeletal muscle physiology, mediating critical adaptive mechanical responses under normal conditions but driving pathology when dysregulated. Understanding the precise regulation of Nox2 and its interaction with other mechanotransduction pathways could inform the development of targeted therapies to reduce oxidative stress and preserve muscle mass in disuse-related conditions.

### 4.2. Therapeutic Strategies Targeting Oxidative Stress and NOX2

#### 4.2.1. Antioxidants

Antioxidants are crucial for counteracting oxidative stress and its detrimental effects on skeletal muscle health [127]. Increased mitochondrial ROS production, endoplasmic reticulum stress, and decreased antioxidant capacity contribute to sarcopenia during unloading. Mitochondrial-targeted antioxidants, such as elamipretide (also known as SS-31) and MitoQ, selectively scavenge mtROS by localizing to the inner mitochondrial membrane, improving electron transport chain (ETC) efficiency and reducing lipid peroxidation [124,147]. EUK-134, a synthetic superoxide dismutase/catalase mimetic, neutralizes cytosolic superoxide and hydrogen peroxide, preventing nNOS delocalization and NOX2 hyperactivation [40]. EUK-134 and SS-31 have shown significant promise in suppressing mitochondrial ROS production, preventing oxidative damage, and preserving muscle mass during mechanical unloading [40,124,148] and aging [149,150]. These compounds reduce mitochondrial H2O2 output and normalize oxidative stress markers such as manganese superoxide dismutase (MnSOD), catalase, and glutathione peroxidase, which are often downregulated under conditions of mechanical unloading [6,134,151,152].

The Lawler Laboratory demonstrated that EUK-134 protects against unloading-induced atrophy by enhancing the antioxidant response, preserving sarcolemmal integrity, and reducing atrogin-1/MuRF1 expression [40]. Natural antioxidants, including salidroside, curcumin, and N-acetylcysteine (NAC), mitigate atrophy via distinct mechanisms. Salidroside suppresses FoxO3a phosphorylation, blocking UPS activation [153], while curcumin upregulates Nrf2, restoring glutathione synthesis and SOD2 activity [154]. NAC, a glutathione precursor, reduces protein carbonylation and rescues satellite-cell proliferation in disuse models [155]. Emerging antioxidants like edaravone, a free-radical scavenger, inhibit lipid peroxidation and NF-κB signaling [156,157,158], while coenzyme Q10 (CoQ10) stabilizes ETC complexes, improving bioenergetics in aged muscle [159,160]. Apha-lipoic acid (ALA) enhances thioredoxin reductase activity and disrupts NOX2 assembly, offering dual antioxidant and anti-inflammatory benefits [161,162].

Despite these promising results, specific interventions have shown limited efficiency in prolonged unloading conditions, and limitations persist. For example, MCAT transgenic mice, which overexpress mitochondrial-targeted antioxidants, exhibit reduced oxidative stress but fail to prevent muscle atrophy after extended unloading periods [163]. Similarly, poor bioavailability and off-target effects hinder natural antioxidants like resveratrol [164,165]. These findings suggest that combination therapies—pairing antioxidants with anti-inflammatory agents or exercise—are essential for sustained benefits. Future strategies may include nanoparticle delivery systems to enhance antioxidant targeting or CRISPR-based Nrf2 activation to amplify endogenous defenses [166,167]. These suggest that antioxidant therapy may need to be combined with other strategies to achieve sustained benefits.

#### 4.2.2. Anti-Inflammatory Agents

The endoplasmic reticulum (ER) plays a pivotal role in protein folding, calcium homeostasis, and lipid synthesis. Under mechanical unloading, oxidative stress disrupts ER homeostasis, triggering the unfolded protein response (UPR) [117]. ER stress sensors—IRE1α, PERK, and ATF6—activate pathways that upregulate chaperones like BiP/GRP78 to restore proteostasis. However, chronic ER stress shifts from adaptive to apoptotic signaling via CHOP activation, promoting caspase-12 cleavage and mitochondrial apoptosis [168,169]. ER stress also amplifies ROS through calcium leakage, which activates mitochondrial ROS production [170,171].

ER stress intersects with proteolytic systems in atrophy. PERK activation induces ATF4, which upregulates atrogin-1 and MuRF1, linking ER stress to UPS-mediated protein degradation [172]. Autophagy is similarly regulated via IRE1α-JNK signaling, which phosphorylates Bcl-2 to release Beclin-1 [173]. Satellite-cell dysfunction under ER stress arises from impaired Notch signaling and myogenic differentiation [25]. Furthermore, ER–mitochondria contact sites facilitate calcium transfer, exacerbating ROS and mitochondrial permeability-transition-pore opening [174]. Therapeutic strategies targeting ER stress, such as chemical chaperones (e.g., TUDCA), mitigate atrophy by reducing UPR activation [23,175].

#### 4.2.3. Controlled Physical Activity and Multimodal Therapies

Controlled physical activity counteracts disuse atrophy by restoring redox balance and mechanotransduction. Resistance exercise upregulates PGC-1α, enhancing mitochondrial biogenesis and antioxidant defenses [176]. Low-intensity aerobic exercise reduces NOX2 activity and IL-6 secretion, while neuromuscular electrical stimulation (NMES) preserves muscle mass in immobilized patients by activating AMPK and inhibiting FoxO3a [177,178].

Nutritional interventions, such as branched-chain amino acids (BCAAs) and vitamin D supplementation, synergize with exercise. BCAAs stimulate mTORC1, suppressing atrogin-1, while vitamin D reduces TNF-α and enhances calcium handling [179,180]. Pharmacological adjuvants like angiotensin receptor blockers (losartan) inhibit NOX2-ROS crosstalk, while β-blockers (carvedilol) reduce catecholamine-driven proteolysis [130,181].

#### 4.2.4. Heat-Shock Proteins

Heat-shock proteins (HSPs) are molecular chaperones pivotal in maintaining proteostasis under stress conditions, with emerging roles in regulating NADPH oxidase (NOX)-mediated reactive oxygen species (ROS) production [40,182]. These proteins, particularly HSP70 and HSP25, protect muscle cells by repairing partially oxidized proteins [183], modulating oxidative stress, and preventing cellular apoptosis [184,185,186]. Research has shown that HSPs are critical for muscle development and regeneration but are often downregulated during periods of mechanical unloading, and directly interact with NOX systems to mitigate ROS overproduction, contributing to atrophy [6,151,152].

HSP70, for instance, mitigates unloading-induced muscle atrophy by binding to cytosolic NOX2 subunits (p47phox, p67phox), inhibiting their assembly and subsequent superoxide generation [6,154]. This interaction stabilizes redox balance, prevents nNOS translation, preserves nitric oxide (NO)-mediated suppression of NOX2 activity and modulates the autophagic flux [187,188,189]. HSP25 stabilizes the actin cytoskeleton, blocking NOX2 translocation to the membrane and reducing ROS-driven proteolysis [190,191].

Interventions that increase HSP70 expression, such as curcumin and fish oil supplementation, have demonstrated significant protective effects by suppressing NOX2-derived ROS and stabilizing sarcolemmal integrity against oxidative stress and muscle atrophy [152,154,182,192,193,194,195,196]. For example, curcumin upregulates HSP70, which in turn inhibits NF-κB activation, a downstream target of NOX2 signaling, thereby reducing atrogin-1/MuRF1 transcription [182].

Recent studies have also highlighted the role of glucose-regulated protein 94 (GRP94), an ER-resident HSP, in preventing nNOS mislocalization, ensuring NO bioavailability to counteract NOX2 hyperactivity, and enhancing muscle resistance to atrophy [194]. It underscores the therapeutic potential of targeting HSP pathways to protect against oxidative stress-induced muscle degradation during unloading and disease states [101,194].

## 5. Interplay Between Focal Adhesions and Mechanical Unloading

Mechanical unloading disrupts the structural and functional integrity of focal adhesions (FAs), pivotal mechanosensors that mediate bidirectional signaling between the extracellular matrix (ECM) and cytoskeleton. Under physiological loading, FAs dynamically adapt to mechanical cues, maintaining muscle homeostasis through integrin-mediated force transmission and the activation of signaling pathways such as focal adhesion kinase (FAK) [78,109]. However, unloading diminishes mechanical tension, destabilizing FA complexes and triggering a cascade of atrophic responses. Integrin α7β1, critical for myotendinous junction stability, is downregulated during unloading, impairing ECM–cytoskeleton coupling and compromising sarcolemmal integrity [53,89]. This disruption reduces FAK autophosphorylation, attenuating downstream survival signals (e.g., PI3K/Akt) while activating proteolytic pathways via FoxO transcription factors [98,111]. These processes are summarized in Figure 3, which illustrates the ECM–integrin–FA–cytoskeleton axis and its dysregulation under mechanical unloading.

FA components such as vinculin and talin, which reinforce integrin–cytoskeletal linkages, exhibit altered localization under unloading, weakening adhesion strength and mechanotransduction [36,66]. Concurrently, costameric structures—specialized FA regions anchoring sarcomeres to the sarcolemma—become disorganized, impairing force transmission and exacerbating sarcolemmal damage [94,99]. As depicted in Figure 3, oxidative stress exacerbates FA instability: NADPH oxidase 2 (NOX2)-derived reactive oxygen species (ROS) oxidize FA proteins, further destabilizing integrin clustering and FA assembly [6,13]. For instance, NOX2 hyperactivation during unloading promotes nNOS delocalization, reducing nitric oxide-mediated suppression of atrogin-1/MuRF1 expression [40,130].

The interplay between FA dysfunction and oxidative stress creates a deleterious feedback loop. Oxidative damage impairs FA repair mechanisms, while FA instability exacerbates mitochondrial ROS overproduction, amplifying proteolysis and inhibiting regeneration [101,133]. Figure 3 highlights key nodes in this cycle, including NOX2/ROS-driven NF-κB activation, FoxO3a-mediated ubiquitin ligase upregulation, and impaired mTOR/Nrf-2 signaling. Therapeutic strategies targeting FA stabilization, such as FAK activation or integrin agonism, alongside NOX2 inhibition, show promise in preserving muscle mass during unloading [6,98]. Additionally, controlled reloading via exercise restores FA signaling by reactivating mechanosensitive pathways like YAP/TAZ and PGC-1α, which enhance antioxidant defenses and mitochondrial biogenesis [24,197]. Thus, understanding FA-ROS crosstalk under mechanical unloading is critical for developing interventions to mitigate disuse-induced atrophy.

## 6. Conclusions

This review has comprehensively addressed the interconnected roles of focal adhesions and oxidative stress within mechanotransduction pathways, emphasizing their critical contribution to skeletal muscle atrophy during mechanical unloading. Focal adhesions and key mechanosensors integrate mechanical signals into biochemical responses essential for maintaining muscle homeostasis. Their disruption under unloading conditions—characterized by integrin α7β1 downregulation, FAK inactivation, and costamere disassembly—impairs force transmission and triggers proteolytic pathways (e.g., FoxO-mediated atrogin-1/MuRF1 upregulation), leading to muscle wasting. In parallel, oxidative stress exacerbates these effects by amplifying reactive oxygen species (ROS) production through mitochondrial dysfunction and NADPH oxidase-2 (NOX2) hyperactivation, particularly in disuse models. This enzyme, while essential for physiological adaptation, drives pathology when dysregulated, disrupting nNOS localization and nitric oxide bioavailability, thereby impairing protein synthesis and muscle regeneration. The interplay between focal-adhesion dysfunction and oxidative stress creates a self-reinforcing cycle: ROS oxidize FA components, destabilizing mechanotransduction, while FA instability exacerbates mitochondrial and NOX2-derived ROS overproduction.

Therapeutic strategies targeting these mechanisms, including NOX2 inhibitors (e.g., peptide-based therapies), mitochondrial-targeted antioxidants (e.g., EUK-134), and FAK-activating compounds, hold promise for preserving muscle function. However, single-target approaches exhibit limited efficacy in chronic unloading, necessitating combinatorial therapies that concurrently stabilize FAs and neutralize ROS. Future research should prioritize the synergistic targeting of FA-redox pathways, such as coupling YAP/TAZ activation with ROS scavengers alongside advanced delivery systems (e.g., nanoparticles) to enhance therapeutic precision. Clinical validation in aging, immobilization, spaceflight models, and personalized approaches informed by omics profiling will be critical to translating mechanistic insights into effective interventions. By elucidating these intricate mechanisms, this review not only clarifies the pathophysiology of disuse atrophy but also charts a translational roadmap to mitigate muscle wasting in diverse clinical and extreme environments.

## Figures and Tables

**Figure 1 ijms-26-02802-f001:**
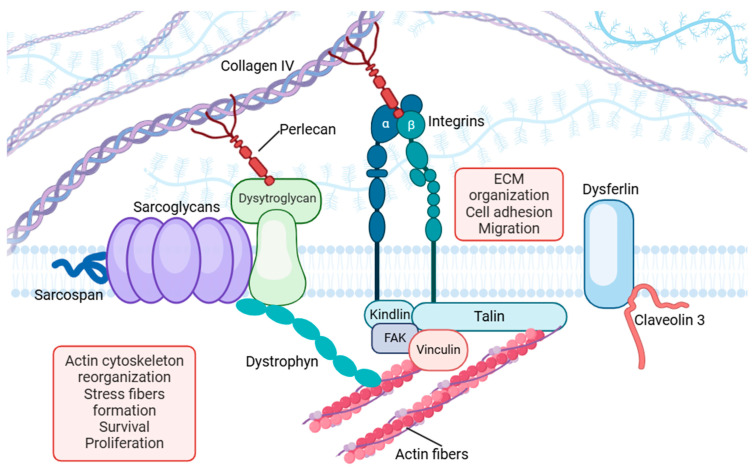
A schematic representation of focal-adhesion elements involved in mechanosensing. Focal adhesions are protein complexes that act as an intermediate between the ECM and the actin skeleton. They play a dual role, influencing both extracellular and intracellular processes. Externally, they regulate ECM organization, cell adhesion, and motility. Internally, they contribute to critical cellular functions such as F-actin polymerization, cell survival, and proliferation. External and internal signaling integration makes focal adhesions essential for coordinating cellular responses to the environment. Figure generated with Biorender.

**Figure 2 ijms-26-02802-f002:**
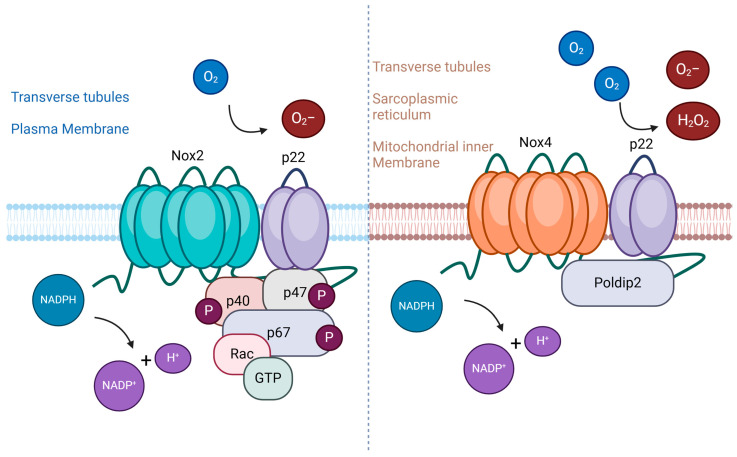
NOX2 and NOX4 as key oxidative-stress mediators. The predominant isoforms of NADPH oxidases are expressed in skeletal muscle. NOX2 is found predominately in the plasma membrane and the transverse tubules and is the primary source of ROS during contractions. NOX4 is mainly found in the transverse tubules, the sarcoplasmic reticulum, and the mitochondrial inner membrane, being involved in mitochondria biogenesis, oxygen sensing, and antioxidant response. Figure generated with Biorender.

**Figure 3 ijms-26-02802-f003:**
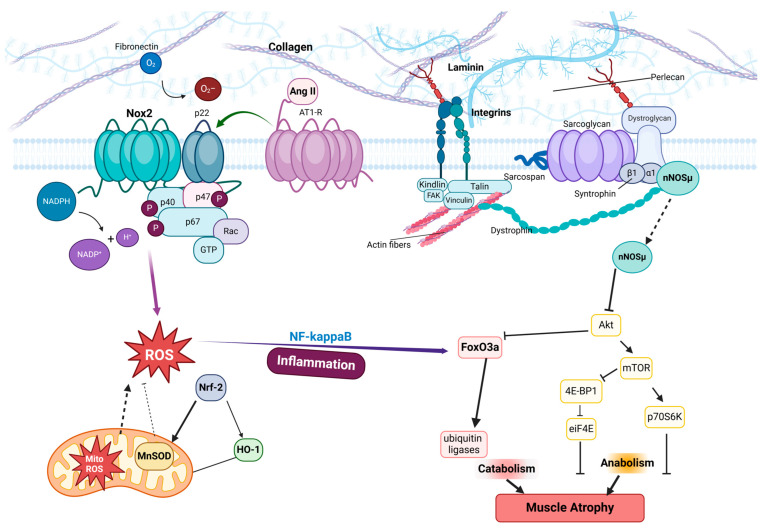
Dysregulation of focal adhesion (FA)–redox crosstalk under mechanical unloading. This schematic illustrates the molecular mechanisms linking FA destabilization and oxidative stress during mechanical unloading. Under physiological loading, integrins (e.g., α7β1) bind extracellular matrix (ECM) components (fibronectin, collagen, laminin, perlecan), stabilizing FA complexes through talin–vinculin interactions that anchor the actin cytoskeleton. Mechanical unloading disrupts integrin–ECM binding, leading to FA disassembly, FAK inactivation, and loss of PI3K/Akt survival signaling. Concomitantly, angiotensin II (Ang II) binding to AT1 receptors activates NOX2 (composed of p22, p47, p67, Rac), generating reactive oxygen species (ROS) that oxidize FA proteins (e.g., the dystrophin–glycoprotein complex) and promote nNOS delocalization. ROS further amplify proteolysis via the NF-κB/FoxO3a-driven upregulation of ubiquitin ligases (MuRF1/atrogin-1) and suppress protein synthesis by inhibiting mTOR/4E-BP1 and Nrf-2-mediated antioxidant defenses (MnSOD, HO-1). The resultant muscle atrophy is exacerbated by a self-reinforcing cycle: FA instability impairs mechanotransduction, while oxidative stress perpetuates mitochondrial ROS overproduction. Therapeutic strategies targeting FAK activation, NOX2 inhibition, or integrin agonism (highlighted in red) aim to restore FA–cytoskeleton coupling and redox balance, mitigating disuse-induced atrophy. Arrows Explanation: Solid arrows (→) indicate activation or stimulation, dashed arrows (⇢) denote indirect interactions or translocation, blunt-ended lines (⊣) represent inhibition or suppression.

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
