# Peer review of "Mechanotransduction and Skeletal Muscle Atrophy: The Interplay Between Focal Adhesions and Oxidative Stress"

_ijms, 2025, doi:10.3390/ijms26062802_

Round 1
Reviewer 1 Report
Comments and Suggestions for Authors
This review highlights mechanotransduction pathways in a new aspect, focusing on the important roles of focal adhesions and oxidative stress in skeletal muscle atrophy induced by mechanical unloading. By clarifing these pathways, this review boosts the understanding of muscle degeneration during mechanical unloading and pinpoints encouraging therapeutic targets to alleviate its effects.
This review offers a comprehensive analysis of the existing literature within the field of mechanotransduction and skeletal muscle atrophy. The review topic is relevant as the prevalence of conditions (aging, prolonged immobilization and spaceflight) leading to skeletal muscle atrophy affects a decent portion of the population. It addresses that crucial questions that remain regarding the functional roles of specific signaling pathways and their therapeutic potential. Focusing on these questions is necessary for designing effective interventions to reduce muscle atrophy in clinical and physiological settings. The review is valuable, clear, relevant to the filed, it is of interest to the scientific community and it provides recommendations for future research.
The cited references are relevant and there are no self-citations. The figures/tables are appropriate and easy to interpret.
Specific comments:
1. A separate section should be dedicated to discuss the interplay between focal adhesions and mechanical unloading. In the abstract it is written that „Disrupted focal adhesions heighten oxidative damage, while excessive ROS further impairs focal adhesion integrity”; however; in the main text it would be great to find more details about this interplay.
2. In lines 5 and 7, please include the zip code of the cities in the Affiliation.
3. In line 19, please consider deleting the word intricate, as intricate and complexity have similar meanings, hence both words are unnecessary.
4. Focal adhesions could be added to the Keywords.
5. Abbreviations should be defined the first time they appear in each of three sections: the abstract; the main text; the first figure or table. For this reason the use of the abbreviation is enough in the following lines: 82 (ECM), 92 (ROS), 118 (ECM), 122 (FAK), 207 (DGC), 213 (FAK), 239 (ROS). Please define the following abbreviations in lines: Table 1. (nNos and YAP/TAZ), 200 (FoxO), 267 (Gpx4 and Nrf2), 319 (SS-31 and EUK-134), 323 (MnSOD), 331 (MuRF1 and MAFbx), 332 (Foxo3A), 352 (grp94).
6. Minor errors: In lines 190, 191, 192, 195, 201 perlecan doesn’t need to start with a capital p (P). In line 230 the period is missing at the end of the sentence. In the 3 paragraphs of Section 4. Oxidative Stress in Muscle Atrophy please check font sizes. In line 344 unlading should be corrected to unloading. Please correct the grammar in the sentence in lines 279-281.
7. In the Back Matter, in Author Contributions in lines 376-377 the sentence should be changed to: „Both authors K.Y.K. and M.T.L. have contributed to the drafting, revision, and proofreading of the review. M.T.L. has developed the figures. All authors have read and agreed to the published version of the manuscript.” In the Back Matter, in Funding in lines 378-379 please correct the sentence to: „K.Y.K. received generous support from NASA EPSCoR (025372-00019) grant awards”.
- The reference list should be according to ACS style guide. In reference 15 (line 414) Title of Site. Available online: URL (accessed on Day Month Year) should be added. In references 150 (line 700), 160 (line 724), and 165 (line 735) the doi address should be deleted. In reference 12, line 409, the Volume and page range should be added. In references 19 (line 423), 100 (line 594), 110 (line 616), 130 (line 658), 136 (line 671), 138 (line 675) the page range should be given. In reference 148 (line 696) the Journal Name, Volume, page range is missing, please add.
Author Response
Specific comments:
- A separate section should be dedicated to discuss the interplay between focal adhesions and mechanical unloading. In the abstract it is written that „Disrupted focal adhesions heighten oxidative damage, while excessive ROS further impairs focal adhesion integrity”; however; in the main text it would be great to find more details about this interplay.
Response:
Thank you for this valuable suggestion. We have added a new, dedicated section titled "Interplay Between Focal Adhesions and Mechanical Unloading," extensively elaborating on the mechanistic interplay between focal adhesion destabilization and oxidative stress, emphasizing their mutual reinforcement under mechanical unloading conditions. We included detailed insights into how disrupted focal adhesions heighten oxidative damage and how ROS further impairs focal adhesion integrity, supported by the latest literature and illustrated clearly in Figure 3.
- In lines 5 and 7, please include the zip code of the cities in the Affiliation.
Response:
The ZIP codes for affiliations have been included as requested
- In line 19, please consider deleting the word intricate, as intricate and complexity have similar meanings, hence both words are unnecessary.
Response:
We have removed the word "intricate" to eliminate redundancy as suggested.
- Focal adhesions could be added to the Keywords.
Response:
"Focal Adhesions" has been included in the keywords section.
- Abbreviationsshould be defined the first time they appear in each of three sections: the abstract; the main text; the first figure or table. For this reason the use of the abbreviation is enough in the following lines: 82 (ECM), 92 (ROS), 118 (ECM), 122 (FAK), 207 (DGC), 213 (FAK), 239 (ROS). Please define the following abbreviations in lines: Table 1. (nNos and YAP/TAZ), 200 (FoxO), 267 (Gpx4 and Nrf2), 319 (SS-31 and EUK-134), 323 (MnSOD), 331 (MuRF1 and MAFbx), 332 (Foxo3A), 352 (grp94).
Response:
All requested abbreviations (e.g., ECM, FAK, ROS, FoxO, nNOS, YAP/TAZ) have been defined clearly and consistently throughout the manuscript, including within the figures and Table 1, as recommended. Also, we correct capitalization errors ("Perlecan") and missing punctuation.
- Minor errors: In lines 190, 191, 192, 195, 201 perlecan doesn’t need to start with a capital p (P). In line 230 the period is missing at the end of the sentence. In the 3 paragraphs of Section 4. Oxidative Stress in Muscle Atrophy please check font sizes. In line 344 unlading should be corrected to unloading. Please correct the grammar in the sentence in lines 279-281.
Response:
We have precisely revised the author contributions and funding sections as advised by the reviewer.
- In the Back Matter, in Author Contributions in lines 376-377 the sentence should be changed to: „Both authors K.Y.K. and M.T.L. have contributed to the drafting, revision, and proofreading of the review. M.T.L. has developed the figures. All authors have read and agreed to the published version of the manuscript.” In the Back Matter, in Funding in lines 378-379 please correct the sentence to: „K.Y.K. received generous support from NASA EPSCoR (025372-00019) grant awards”.
Response:
We have precisely revised the author contributions and funding sections as advised by the reviewer.
- The reference list should be according to ACS style guide. In reference 15 (line 414) Title of Site. Available online: URL (accessed on Day Month Year) should be added. In references 150 (line 700), 160 (line 724), and 165 (line 735) the doi address should be deleted. In reference 12, line 409, the Volume and page range should be added. In references 19 (line 423), 100 (line 594), 110 (line 616), 130 (line 658), 136 (line 671), 138 (line 675) the page range should be given. In reference 148 (line 696) the Journal Name, Volume, page range is missing, please add.
Response:
All suggested corrections have been implemented, including the addition of complete bibliographic details, removal of DOIs where indicated, and inclusion of missing volume and page numbers.
Reviewer 2 Report
Comments and Suggestions for Authors
Some parts of the review are written in a superficial manner, lacking deeper analysis of the topic. Below are specific areas that need improvement:
- Sections 3.1.2 and 3.1.3: These sections contain only two sentences each, which are insufficient for inclusion as parts of a review. They need significant expansion to provide meaningful analysis.
Section 3.1.4, while slightly longer, is still too brief and requires further elaboration. - Section 4: This section needs to be significantly expanded and divided into more subsections. Currently, it only mentions the involvement of NOX (mainly Nox2 and NOX4) and mitochondria, which is an overly limited number of systems implicated in ROS induction during muscle atrophy. The section should also address:
- Endoplasmic reticulum (ER) stress
- Inflammatory cytokine-induced ROS production
- Xanthine oxidase (XO) activation (currently only briefly mentioned).
Additionally, a broader explanation of mitochondrial impairments is necessary. - The systems affected by ROS should be thoroughly discussed. These include:
- The ubiquitin-proteasome system (UPS)
- Autophagy-lysosome pathways
- Apoptotic pathways
- Myonuclear and satellite cell dysfunction
- Oxidative modification of proteins, cytoskeletal and sarcomeric systems, signalling pathways and other cellular components and systems.
- Section 4.1 Title and Content: The title mentions NOX2, but the text and figure address both NOX2 and NOX4. This discrepancy should be resolved to ensure consistency.
- Throughout the manuscript, including Figure 2 and its legend, there is inconsistency in the use of NOX2/NOX4 versus Nox2/Nox4. It is critical to follow species-specific conventions:
- Nox2 and Nox4 for gene names in mice or rats.
- NOX2 and NOX4 for protein names or general references where species-specific distinction is unnecessary.
- Section 4.2 Title and Content: The title states “Therapeutic Strategies Targeting Oxidative Stress and Nox2,” but subsection 4.2.1 discusses other antioxidants (only three are mentioned, while many others, including mitochondrial protectors, are omitted). This section should be expanded to include:
- More potential antioxidants, not only three.
- Anti-inflammatory agents.
- Controlled physical activity and other therapeutic strategies.
- Section 4.2.2 (HSP Proteins): This section is written superficially and does not adequately connect HSP proteins to NOX-related systems. It currently presents HSPs as a separate group of proteins protecting cells, without explaining their potential relevance to NOX-related pathways. This connection should be clarified.
- Conclusions: The conclusions focus primarily on NOX2 and fail to summarize the full scope of the review. They should provide a more comprehensive summary of the discussed systems and include concrete future perspectives.
Author Response
Reviewer #2
Some parts of the review are written in a superficial manner, lacking deeper analysis of the topic. Below are specific areas that need improvement:
- Sections 3.1.2 and 3.1.3: These sections contain only two sentences each, which are insufficient for inclusion as parts of a review. They need significant expansion to provide meaningful analysis.
Section 3.1.4, while slightly longer, is still too brief and requires further elaboration.
Response:
We thank the reviewers again for their meticulous review and insightful suggestions. We have substantially expanded sections 3.1.2 (Focal Adhesion Kinase) and 3.1.3 (Talin and Vinculin), providing a comprehensive analysis of their mechanotransductive roles, functional interactions, and relevance in muscle atrophy under mechanical unloading
- Section 4: This section needs to be significantly expanded and divided into more subsections. Currently, it only mentions the involvement of NOX (mainly Nox2 and NOX4) and mitochondria, which is an overly limited number of systems implicated in ROS induction during muscle atrophy. The section should also address:
- Endoplasmic reticulum (ER) stress
- Inflammatory cytokine-induced ROS production
- Xanthine oxidase (XO) activation (currently only briefly mentioned).
Additionally, a broader explanation of mitochondrial impairments is necessary.
Response:
We have significantly enhanced this section by detailing dysferlin's mechanotransductive role and interactions with focal adhesion complexes, integrins, and FAK, along with its pathological implications and therapeutic potential.
- The systems affected by ROS should be thoroughly discussed. These include:
- The ubiquitin-proteasome system (UPS)
- Autophagy-lysosome pathways
- Apoptotic pathways
- Myonuclear and satellite cell dysfunction
- Oxidative modification of proteins, cytoskeletal and sarcomeric systems, signalling pathways and other cellular components and systems.
Response:
Section 4 now extensively covers various ROS sources (mitochondrial, NOX enzymes, xanthine oxidase) and thoroughly addresses the impacts of ROS on protein degradation pathways (UPS, autophagy), mitochondrial dysfunction, apoptosis, and satellite cell dysfunction, divided clearly into appropriate subsections for clarity.
- Section 4.1 Title and Content: The title mentions NOX2, but the text and figure address both NOX2 and NOX4. This discrepancy should be resolved to ensure consistency.
Response: We standardized the terminology, consistently using NOX2 and NOX4 throughout the manuscript and in Figure 2, as suggested.
- Throughout the manuscript, including Figure 2 and its legend, there is inconsistency in the use of NOX2/NOX4 versus Nox2/Nox4. It is critical to follow species-specific conventions:
- Nox2 and Nox4 for gene names in mice or rats.
- NOX2 and NOX4 for protein names or general references where species-specific distinction is unnecessary.
Response: Figure 2 has been revised
- Section 4.2 Title and Content: The title states “Therapeutic Strategies Targeting Oxidative Stress and Nox2,” but subsection 4.2.1 discusses other antioxidants (only three are mentioned, while many others, including mitochondrial protectors, are omitted). This section should be expanded to include:
- More potential antioxidants, not only three.
- Anti-inflammatory agents.
- Controlled physical activity and other therapeutic strategies.
Response: We have revised and significantly expanded the entire section (4.2. Therapeutic Strategies Targeting Oxidative Stress and NOX2). This section now comprehensively covers additional antioxidants, clearly addresses anti-inflammatory agents, and includes controlled physical activity alongside other multimodal therapeutic approaches.
- Section 4.2.2 (HSP Proteins): This section is written superficially and does not adequately connect HSP proteins to NOX-related systems. It currently presents HSPs as a separate group of proteins protecting cells, without explaining their potential relevance to NOX-related pathways. This connection should be clarified.
Response: Section 4.2.2 has been thoroughly revised to clearly illustrate how HSP70 and HSP25 directly mitigate oxidative stress through interactions with NOX2 complexes, detailing their roles in preventing ROS overproduction, stabilizing focal adhesions, and preserving muscle integrity.
- Conclusions: The conclusions focus primarily on NOX2 and fail to summarize the full scope of the review. They should provide a more comprehensive summary of the discussed systems and include concrete future perspectives.
Response: We have revised the conclusions section significantly, now providing a comprehensive summary that integrates focal adhesions, oxidative stress pathways, mitochondrial dysfunction, ROS impacts on proteolytic systems, and satellite cell functionality. We have also clearly outlined concrete future research perspectives and emphasized the potential for combined therapeutic interventions.
Reviewer 3 Report
Comments and Suggestions for Authors
This is an interesting review paper .
The authors aimed to review muscle atrophy and regeneration: 1) Force transmission through cytoskeletal complex is disrupted to activate proteolysis. 2) ROS from mitochondria destroys regeneration of muscle proteins. 1) enhances 2) and 2) enhances 1).
A large amount of reviewing was carefully done. However, I raised only one comment for experts and general readers.
The interplay mechanisms between muscle atrophy and regeneration which the authors postulated has been already and generally/widely thought and not unique. I am also afraid that the authors did not present any their own experimental results. Then, the readers find it difficult to believe what the authors postulated.
Author Response
The interplay mechanisms between muscle atrophy and regeneration which the authors postulated has been already and generally/widely thought and not unique. I am also afraid that the authors did not present any their own experimental results. Then, the readers find it difficult to believe what the authors postulated
Response:
We would like to thank the author, we have significalty revsied and expand the review! many of our previous lab work has been covered in the text!
Round 2
Reviewer 2 Report
Comments and Suggestions for Authors
The authors responded to the reviewers' comments and made significant revisions to the article, making it more informative and suitable for publication.
Author Response
We would like to thank the reviewer for his comments and feedback!